# Developing a real-time hand-gesture recognition technique for wheelchair control

**Md Rafiul Huda, Md. Liakot Ali**[ID]*, **Muhammad Sheikh Sadi**[ID]

Institute of Information and Communication Technology (IICT), Bangladesh University of Engineering and Technology (BUET), Dhaka, Bangladesh

* liakot@iict.buet.ac.bd

## Abstract

At present, significant number of people in the world are having motor disabilities. They need to use wheelchair for performing regular movements and activities. However, there are a lot of issues and challenges in using the conventional wheelchairs. Safe navigation, independent mobility and low cost are the key issues for wheelchair users. This paper presents a new mathematical model for an efficient and real-time hand gesture recognition technique for a smart wheelchair control system, in line with contemporary technologies. The model is developed using the positions of the significant hand landmarks, distances among them and some critical thresholds which are determined by good number of samples of hands. The proposed method offers more users' flexibility through less hand movement in operation of the wheelchair. The model is tested for hands of different sizes, irrespective of any background in indoor or outdoor under sunlight. The experimental study demonstrates that it outperforms the existing methods in terms of success rate with good performance metrics.

## 1. Introduction

One of the most important desire of human life is his/her free and independent mobility. Every human life deserves the right to move around freely. According to the World Health Organization (WHO), approximately 15% of the world's population has a disability with problems in their body's structure or function or face difficulties in performing action in daily life [1]. Out of these numbers, 131.8 million people due to their diseases or disabilities like hemiplegia, quadriplegia, paraplegia, brain injury, cerebral palsy [2], broken leg, and more (or 1.85% of the global population) require a wheelchair to go around daily [3]. It is among the most widely used assistive technologies that gives support for independent navigation to the people with disabilities and its demand is raised to make their life easier [4]. Manual wheelchairs are fulfilling the necessity for many years. But it is very difficult for the people with severe disabilities to run a wheelchair manually by their own. Hence, they require human assistance for safe navigation with wheelchair. Also, the long-time use of manual wheelchair may affect the user's health and muscles [5]. So powered wheelchair is developed to mitigate these problems [6]. In this era of ICT, an electronic system can be integrated to a wheelchair to make it smarter and more intelligent. The vast majority of developed systems use technologies like Artificial Intelligence (AI), the Human-Machine Interface (HMI), Brain-Computer

**Data availability statement:** In this work, no separate data set was used or required for any

training or testing, as a mathematical model is developed and gesture recognition is done live based on that mathematical model. For the hand detection part only, MediaPipe framework is used, which contains a pre-trained hand landmark model that employs direct coordinate prediction or regression to precisely locate 21 3D coordinates representing hand knuckles. Then, for gesture recognition, a mathematical model is developed using distances among hand landmarks and some thresholds, which is the major part of the proposed methodology and requires no data set. In this part, continuous measurement of distances among hand landmarks on actual live hand is performed. Upon satisfying the different logical conditions as mentioned in the manuscript, different gestures are recognized and relevant wheelchair operations are performed. Accuracy was measured from outputs by capturing such live hand samples multiple times for each gesture accordingly. However, extracted data and outputs from scanned image have been added in supporting information files.

**Funding:** The author(s) received no specific funding for this work.

**Competing interests:** The authors have declared that no competing interests exist.

Interface (BCI) [7], Motion Sensors, and Computer Vision [8-9]. But, in order to make the wheelchair more suitable for disabled people specially who have low energy and lower control of hands, it needs to be more user-flexible and easily controllable. Also, the cost should be minimized at the same time to make it accessible to greater range of population.

Gestures are an easy yet powerful means to express emotions and interact with others. Touch, joystick control, movement, and other methods of physical contact are additional forms of communication. In order to use things like contemporary smartphones, physical contact is required. Joystick operations generally require more energy and more movements of hands and fingers. But, a significant percentage of the population is incapable to engage in physical activity and does not possess the strength necessary to operate a typical joystick. Gesture is an easier way to perform wheelchair operations for disabled people who face difficulties in moving their hands properly. Consequently, gesture communication is a practical technique that is currently receiving greater attention [10]. Gestures can easily be integrated into advanced wheelchair functionalities, making wheelchairs easier to control, through the use of computer vision, camera, and AI-based technologies like deep machine learning [11].

Several researches have been performed utilizing the gestures by various organs, i.e., hand gestures, head movements and eye gaze. Recent innovations in this field include depth cameras such as Microsoft's Kinect [12] and gesture recognition models with such depth cameras have been developed by a number of researchers [13-14]. But these cameras are more expensive than the typical RGB camera and the accuracy varies with background complexity [19]. Additionally, using an RGB camera to achieve better results can be challenging due to the lightning of the environment. Consequently, it is challenging in this industry to recognize gestures using RGB camera and inexpensive equipment. Another researcher has designed a hand gesture recognition model utilizing Haar-cascade classifier [15], skin segmentation [16] and 2D CNN. However, this model doesn't perform well at outdoor under daylight or if any part of the background coincides with the skin color of the hand. Hence, it is still now a challenging issue to build an efficient gesture recognition technique for wheelchair control that will consider users' flexibility, high accuracy, and environment independency.

In this paper, we have proposed a unique mathematical approach for hand gesture recognition in real-time for the control of a smart wheelchair using only finger movement in front of an RGB camera. The user will be able to control the wheelchair by simple natural movements of fingers comfortably. The entire process is divided into three main steps: Hand detection and hand tracking using MediaPipe Hands [17], hand landmarks extraction, and gestures recognition using a unique mathematical model. The proposed model is distinctive in that it compares the distances between key hand landmarks to certain thresholds that are determined earlier using the locations of the major hand landmarks and the minimum, maximum, and average normalized distances among the major landmarks in real time. Some important thresholds have been determined and some logical conditions have been developed to recognize different gestures. After these logical conditions have been satisfied, different control operations are performed based on different gestures. In contrast to the conventional joystick control technique, the proposed technique will allow a smart wheelchair system to be easily operated by disabled people by using only natural finger gestures and movements which makes it more flexible and easily controllable. It requires less energy and effort of the users. The proposed technique utilizes a simple RGB camera (1 Mega Pixel, 30 FPS, and USB cable connection) to detect and recognize gestures and movements and no joystick operations or separate sensors are required.

The three gestures which are taken into consideration in this research stand in for three modes of wheelchair operation: 'Drive', 'Stop' and 'Horn'. When in driving mode, a specific finger is tracked and wheelchair's directional movement is performed accordingly as per the

direction of that finger. The wheelchair's movement is stopped using the 'Stop' mode, and any obstacles are alerted to using the Horn mode.

Contributions of this research are -

- A unique mathematical model has been designed utilizing hand landmarks and some thresholds for hand gesture recognition

- An efficient, flexible gesture recognition system for the control of a wheelchair has been proposed which requires only movement of fingers

- A better gesture recognition system for wheelchair control has been proposed focusing these aspects all together: users' flexibility, high accuracy, and environment (indoor and outdoor) independency.

The rest of the paper is summarized as follows. Section 2 represents a study of related works which are developed by various researchers. After that, Section 3 elaborates the proposed technique of whole gesture recognition process used in this research. Section 4 provides a detailed analysis of the experiment performed using the proposed technique, including comparison with other similar developed systems. Section 5 concludes the paper.

## 2. Related work

The features of smart wheelchairs are continuously being improved by researchers and developers. Over the past few years, a number of smart wheelchair systems with various control mechanisms have been proposed. We have discussed a few of these techniques as they pertain to the suggested system for consistency's sake.

For instance, a Bluetooth-based recognition system using hand or finger movements was presented by Megalingam et al. [18] using an android application running on an intelligent device. The user must navigate the touchscreen of the system with their hands or fingers. The tablet display must be bigger so that the entire palm fits on the tablet. However, it might be challenging for those who are unable to use that particular system.

An intelligent wheelchair system that is controlled by hand gestures was suggested by Gao et al. [19]. Real-time gesture tracking is accomplished using a powerful laptop and the Kinect depth camera. It acquires the input signal based on the gesture. Since the user needs to raise his/her hands to perform gestures, the proposed system is not always user-friendly for the disabled or elderly people who cannot move their hands properly. Depth cameras are also expensive compared to simple RGB camera.

Another wheelchair control system utilizing the iris movement was created by Desai et al. [20], allowing the wheelchair user to control the wheelchair by moving their eye's iris. MATLAB programming embedded within this system was used to implement it. For detecting eye movements, a HD camera was used with the main idea being to transmit the images of the eye to the embedded system. Circular Hough Transform (CHT) algorithm was considered for the eye detection is the which aims to recognize circular patterns. However, this proposed system causes complications in real-time.

Oliver et al. [21] devised a method to operate powered wheelchairs by using a gesture-controlled joystick manipulator. The proposed joystick manipulator utilized a body-mounted accelerometer, used for gesture control through motion detection. The detected movements made by the user are then processed by the Arduino microprocessor mounted in the control unit of the joystick. The joystick control unit then mechanically manipulates the powered wheelchairs installed two-axis proportional joystick with two servo motors, maneuvering the powered chair in the desired direction. For users with dexterity issues, wearing an

accelerometer band and performing gestures with hand movement may not be suitable always.

Kutbi et al. [22] have suggested a head movement tracking-based wheelchair control model. In this work, the user's head-motion is detected through a head-mounted, outward-facing egocentric camera. The wheelchair was modeled using the TI-TAN18CS, the command processor was an Arduino Mega, and the framework was Robot OS (ROS). Head-motion is estimated from the difference of the pose of the head relative to the wheelchair frame in two consecutive frames. The system's performance was around 85.7%. The system's price is still high, and using a head-mounted egocentric camera is not user-friendly.

Mahmud et al. [23] have created a multi-modal wheelchair control mechanism. This system utilized a raspberry pi, an RGB camera, a modified VGG-8 model, and an accelerometer for tracking head movement, hands, and eye gaze, respectively. The performance success rate varies around 90%. The two drawbacks are that the eye gaze detection mechanisms are not user-friendly and that tracking sensors must be attached to the user's body.

Tejonidhi et al. [24] have proposed an eye-pupil tracking-based wheelchair movement system. For the purpose of identifying the eyeballs in RGB photographs, this system utilized a Philips microcontroller and the Viola-Jones MATLAB algorithm. The performance varies from 70% to 90%, and the detection procedure is crucial for real-time applications. As a result, it cannot be used in a real-world situation.

Sadi et al. [25] designed a wheelchair system which is controlled by hand gestures in real time, where the authors used YCrCb skin segmentation technique, Haar-cascade classifier, KCF tracking algorithm, and a 2D CNN model. The proposed 2D CNN model contains three convolutional and three max-pooling layers. However, this system is not very reliable and doesn't perform well under bright light or when the color of the background beneath the hand and the skin tone becomes similar.

An overview of associated works with their major attributes is presented in Table 1.

## 3. Proposed technique and its implementation

The proposed gesture recognition technique has been developed considering the highest user flexibility for disabled persons so that wheelchair can be controlled by using minimum movements of hand or fingers. Also, the limitations of related work are also kept in mind during designing the methodology. The proposed gesture recognition technique has three fundamental steps: hand detection and tracking, hand landmarks extraction, and the recognition of hand gestures using an effective mathematical model, as shown in Fig 1. The mathematical model will be used for gesture recognition by developing some logical conditions and then different control operations will be performed as per different gestures. It has been created using some crucial calculations and logical conditions. The model was created using the distances between the major hand landmarks and a few threshold values. These thresholds are determined at the beginning of gesture recognition for once from a number of real hand samples. The buildings blocks are discussed in detail in this paper.

### 3.1. Hand detection and hand tracking

The term 'Hand detection' refers to the identification of the hand area within the frame of an image. This is the primary step in recognizing hand movements. During this stage, a low-cost RGB camera is used to capture the image of the user's hand from top position. To create the scenario of the wheelchair, hand is kept on a handle and the RGB camera is placed on top of the hand. Then hand is detected and its motion is tracked accordingly. Tracking involves

**Table 1. An overview of related work.**

| Existing Technique | Functionality | Limitations | Major Equipment | Accuracy |
|---|---|---|---|---|
| Gao et al [19]. | Hand gesture recognition | Requires hand raising, background complexity | Microsoft Kinect Camera, highly configured laptop | 10-100% (Depends on background complexity) |
| Oliver et al [21]. | Hand movement detection | Requires wearing hand band | Accelerometer, Joystick Manipulator | Actual accuracy is not measured |
| Sadi et al [25]. | Hand gesture recognition | Requires finger movement, doesn't work well under daylight or if the background and skin tone match | RGB Camera, Raspberry Pi | 97.14% |
| Megalingam et al [18]. | Finger movement detection | The user must navigate the touchscreen with their hands or fingers and know how to operate the application. | Arduino Uno, HC05, Bluetooth, Android device | 98% |
| Desai et al [20]. | Eye movement detection | It's a preliminary work to model the control of a wheelchair, creates complication in real-time | HD Camera, Raspberry pi (planned) | Actual accuracy is not measured |
| Kutbi et al [22]. | Head movement Tracking | User's head-motion is detected through a head-mounted egocentric camera, which is not user-friendly. | An egocentric camera, TI-TAN18CS, Arduino Mega | Accuracy not measured directly. Navigation time, number of collisions, survey comments were measured for a route in different modes. |
| Mahmud et al [23]. | Multi-modal wheelchair control mechanism | It's integration of a joy stick with hand gesture, eye gaze and head movement tracker. Not user-friendly, tracking sensors must be attached to the user's body | Raspberry pi, RGB camera and an accelerometer | Joystick: 94% Hand gesture: 92% Eye gaze: 86% Head movement: 90% Multiple: 94% |
| Tejonidhi et al [24]. | Eye pupil's movement detection | Eye pupil is tracked by a camera and the system glows LEDs based on eye movements. The detection procedure is crucial for real-time applications | Web camera, Philips microcontroller | 70% to 90% |

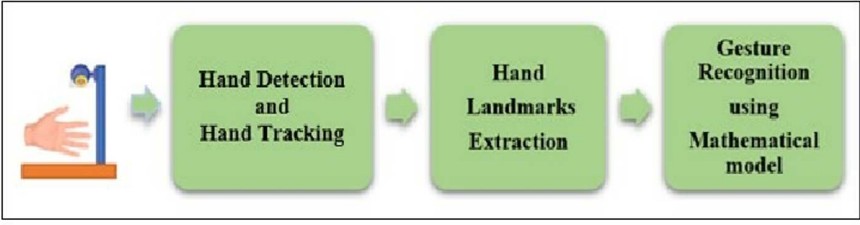

**Fig 1. Steps of the proposed hand gesture recognition technique.**

monitoring the movement of an object within a predetermined visual frame. To facilitate the development of AI-based applications, various frameworks for hand gesture detection have been recently established. One such framework is MediaPipe, which offers multiple options including Instant Motion Tracking, Hand Detection, Iris, Pose, Face Recognition, Object Detection, and more. In this work, the suggested technique utilizes the MediaPipe Hands [26] solution, a high-fidelity technology for tracking hands and fingers, to achieve better performance and resilience. This solution employs a machine learning pipeline composed of multiple interconnected models to infer 21 3D landmarks of a hand from just a single frame. The captured image is required to be transformed from the BGR format (read by OpenCV) to the RGB format before the processing, since the ML pipeline expects RGB image as input. The hand land marker model bundle comprises a palm detection model and a hand landmarks detection model. The Palm detection model traces hands within the input image, and the hand landmarks detection model detects specific hand landmarks on the cropped hand image

defined by the palm detection model. Initially, a hand bounding box is created based on palm detection using a model. The cropped image region defined by the palm detector is then used to generate accurate 3D hand keypoints using a hand landmark model. This hand landmark model employs direct coordinate prediction or regression to precisely locate 21 3D coordinates representing hand knuckles within the detected hand regions. This pre-trained model has been trained using approximately 30,000 real-world photos manually categorized with precise 3D coordinates, as depicted in Fig 2. To use the technique, the user simply needs to place their hand within a specific boundary under the RGB camera. Once the code is executed, the hand and its 21 landmark locations are continuously perceived and tracked, regardless of different gestures or hand sizes. It can detect hands and the landmarks of any size at any distance from the camera. That is why it's very robust and efficient. After the detection, the positions of the hand landmarks will be utilized in next steps.

### 3.2. Hand landmarks extraction

Once the input image of the hand is captured and processed, the coordinates of 21 important hand landmarks are haul out and organized into a matrix. Hand landmarks are the few major points of the hand consisting of fingertip, wrist and joints of the fingers. The coordinates of these landmarks are typically represented as normalized values (x, y, z) format. Here 'x' corresponds to the x-coordinate of the landmark standardized to the range of [0.0, 1.0] based on the width of the image. Similarly, 'y' represents the y-coordinate of the landmark normalized to the range of [0.0, 1.0] based on the height of the image. The z-coordinate signifies the depth of the landmark and is also normalized to a scale that is roughly proportional to 'x'. The reference point for 'z' is the wrist. These coordinates are detected during hand detection irrespective of size of hand. Hand is placed on an area and image is taken continuously by RGB camera from the top position. Since the landmark coordinates are originally normalized, they need to be transformed back to the actual scale. This is achieved by multiplying the coordinates by the image width and image height. The resulting data is then stored in a separate matrix, which is utilized in the suggested model in further calculation. Fig 2 illustrates the major 21 hand landmarks of a hand [12]. For wrist of the hand, 1 landmark is detected. For thumb, index finger, middle finger, ring finger and pinky finger, 4 landmarks are detected for each finger. For any gesture, the landmarks' positions are changed and updated in matrix automatically. These hand landmarks are the important data of hand and these are used in later steps to build the proposed mathematical model.

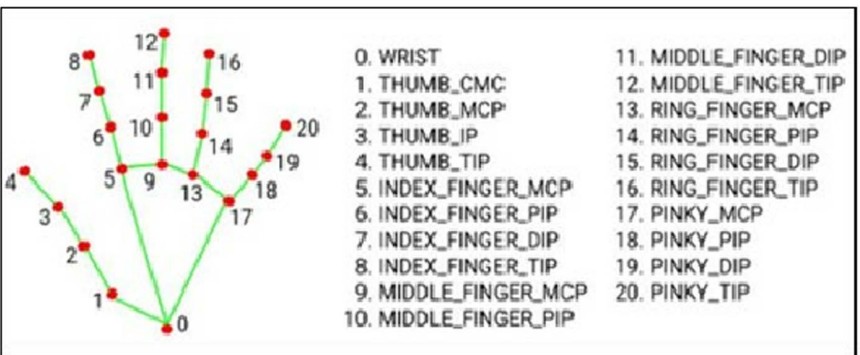

**Fig 2. Major hand landmarks or the key-points.**

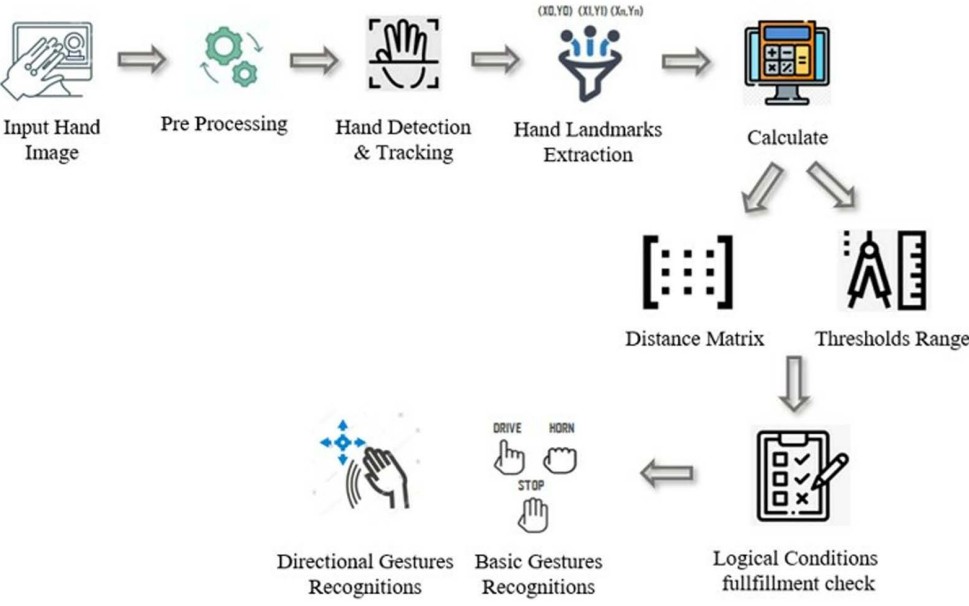

**Fig 3. Architectural elements of the proposed hand gesture recognition technique.**

## 3.3. Hand gestures recognition via mathematical model

The suggested procedure in this research presents a simple design structure comprising hand detection, landmarks abstraction, the distance matrix composition, and employment of a mathematical model for recognizing the hand gestures, as illustrated in Fig 3. Once the real (x, y) coordinates of the 21 landmarks are obtained, the distances from each landmark to the remaining 20 key-points are computed using the distance calculation formula. These distances are then stored in a hand distance matrix in a pattern. To ensure the detection works effectively at any hand-camera distance, all distances are normalized relative to the hand width. The calculated distance amid the key-points 5 and 17 is defined as the hand width (Fig 2). Therefore, the distances are normalized by this hand width accordingly. This normalization allows proposed model's logical conditions to be applicable for hand of any size. Subsequently, by applying multiple logical conditions on the normalized distances with the range of three crucial thresholds determined earlier, different gesture patterns are recognized and decisions regarding wheelchair control operations are made. Each specific gesture will be used to perform a specific control operation. Gestures are changed using the movement of fingers only which has made the proposed model more flexible and suitable for all types of disables persons.

Initially, the proposed technique focuses on three specific gestures for three fundamental control operations: 'Drive', 'Stop', and 'Horn'. 'Drive' gesture is used for moving the wheelchair in any direction. 'Stop' gesture is used to stop the movement of the wheelchair. And the 'Horn' gesture is applied to alert anyone or any vehicle in front of the wheelchair. Additionally, within the 'Drive' gesture category, four directional gestures are implemented, namely 'Drive Backward', 'Drive Right', and 'Drive Left'. The 'Stop' gesture serves as the initial gesture to halt the wheelchair in any situation or condition. On the other hand, the 'Horn' gesture is utilized to indicate or signal the surrounding environment of wheelchair. As soon as the 'Drive' gesture is recognized, the system transitions into the drive mode, enabling the execution of

**Class 1: Drive**  **Class 2: Horn**  **Class 3:  Stop**

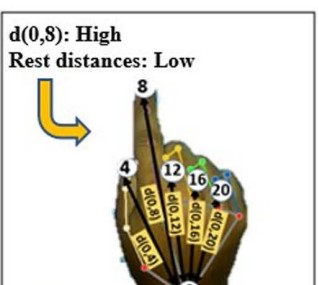 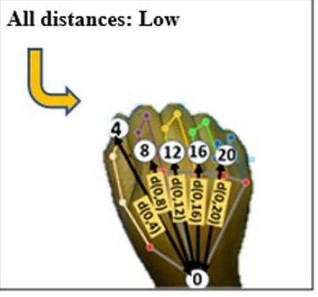 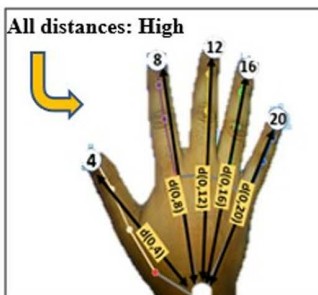

**Fig 4.  Major classes of hand gestures considered in the proposed technique.**

movement operations. Fig 4 illustrates the basic classes of hand gestures and the observations which are used to perform the three basic control operations of a wheelchair in this paper.

During the implementation of the proposed 'Stop' gesture in this work, it was seen that all fingertips maintain the maximum detachment from the hand wrist (keypoint 0). Based on this concept, conditions are prepared on hand landmark distances with some thresholds and the 'Stop' gesture is implemented. When all the distances between keypoint 0 and the fingertips (keypoints 8, 12, 16, and 20) are higher than a specific threshold at a given time, and a certain distance threshold is maintained among the fingertips themselves (keypoints 8, 12, 16, and 20), then the gesture will be recognized as 'Stop'. To enhance the efficiency and flexibility of the model, a range of maximum and minimum values is determined for these thresholds instead of using fixed values. These thresholds are determined at the beginning of the hand detection process from a good number of samples in two steps. The mechanism of determining these thresholds are discussed afterwards in this paper. Mainly three thresholds are used in this work to develop logical conditions need to be satisfied for each gesture.

Similarly, for the 'Horn' gesture, it was observed that the fingertips stay closest to the wrist of the hand (keypoint 0). Again, when the detachments measured amid the keypoint 0 to keypoints 8, 12, 16, and 20 are lower than a specific threshold, then the gesticulation is detected as 'Horn'. A range is considered for this threshold as well, allowing flexibility to the user.

To identify the 'Drive' mode gesture, it is observed that, only the apex of the index finger (keypoint 8) maintains a greater distance from the wrist of the hand (keypoint 0), while the other keypoints of the fingertips stay close to the wrist. Based on this concept, the 'Drive' gesticulation is implemented. For this gesture, also a threshold range is used which made the proposed model more efficient and flexible for the disabled people. If the distance between keypiont 0 and keypoint 8 falls in a specific range of threshold and all other distances are less than another threshold, then 'Drive' gesture is detected. If any disabled person cannot stretch his index finger fully, he or she will also able to operate this system due to considering range of thresholds. During applying the range value it is also considered that it should not be too much flexible which can make the system confused and detect the gestures wrongly.

The logical conditions for recognizing this gesture are outlined in Table 2. Here, d(a,b) reflects the normalized distance amid the keypoint 'a'$(x_1, y_1)$ and keypoint 'b'$(x_2, y_2)$ of the hand. The formula used to determine this distance between the keypoints is illustrated in (1).

$$d(a,b) = [(x_2 - x_1)^2 + (y_2 - y_1)^2]^{1/2} \qquad (1)$$

**Table 2. Conditions for basic gestures recognition in the proposed technique.**

| Gestures | Distance parameters | Required conditions to meet |
|---|---|---|
| **Drive** | $d(0,8)$ | $Thld1_{min} < d(0,8) < Thld1_{max}$ |
| | $d(0,12)$ | $Thld2_{min} < d(0,12) < Thld2_{max}$ |
| | $d(0,16)$ | $Thld2_{min} < d(0,16) < Thld2_{max}$ |
| | $d(0,20)$ | $Thld2_{min} < d(0,20) < Thld2_{max}$ |
| | $d(0,4)$ | $d(0,4) < Thld1$ |
| **Stop** | $d(0,8)$ | $d(0,8) > Thld1$ |
| | $d(0,12)$ | $d(0,12) > Thld1$ |
| | $d(0,16)$ | $d(0,16) > Thld1$ |
| | $d(0,20)$ | $d(0,20) > Thld2$ |
| | $d(0,4)$ | $d(0,4) > Thld2$ |
| | $d(4,8)$ | $Thld3_{min} < d(4,8) < Thld3_{max}$ |
| | $d(8,12)$ | $Thld3_{min} < d(8,12) < Thld3_{max}$ |
| | $d(12,16)$ | $Thld3_{min} < d(12,16) < Thld3_{max}$ |
| | $d(16,20)$ | $Thld3_{min} < d(16,20) < Thld3_{max}$ |
| **Horn** | $d(0,8)$ | $d(0,8) < Thld1_{min}$ |
| | $d(0,12)$ | $d(0,12) < Thld2$ |
| | $d(0,16)$ | $d(0,16) < Thld2$ |
| | $d(0,20)$ | $d(0,20) < Thld2$ |

In Table 2, 1st column represents the name of different basic gestures which are used in this thesis. 2nd column represents the distances among the major landmarks or keypoints of hand. For example, d(0,8) represents the normalized distance between keypoint 0 and keypoint 8 of the hand, as marked in Fig 2. The right most columns represents the logical equations which need to be met for particular gesture recognition. Some distances are required to be within a range of thresholds, some are required to be less than a particular threshold whereas some distance values should be bigger than a particular threshold for particular gesture recognition.

Here, Thld1, Thld2 and Thld3 are the three thresholds which were determined and utilized in this work. The range of these thresholds are determined at the start of hand detection for once from a good number of samples. During the gesture recognition process, the distances are compared with these thresholds and gesture is recognized if all the conditions are met. For example, in Table 2, for 'Drive' gesture, 5 distances and 4 conditions are considered, and all of the 4 conditions on those distances are needed to be satisfied together to recognize that particular gesture. Otherwise gesture will not be recognized.

For Thld1, maximum and minimum range is determined from distances between the hand wrist and the fingertips at the time when all the fingertips remain at the maximum distance from wrist (fingers are opened). The minimum and maximum values are fetched from the distance matrix. Thld2 is an additional threshold determined from the distances between wrist and finger tips at the time when fingertips maintain the minimum distance from the wrist. The last threshold, Thld3 is the distances between any two consecutive fingertips. The system will ask user for giving these two specific gestures as input to calculate threshold ranges. For all three thresholds, 100 of live hand images are taken by RGB camera. Each of the distance

from wrist to five fingertips are different naturally. Also, the gaps between any two fingers are also not the same. All these required distances are measured and the maximum, minimum and mean value is determined for each threshold from the distance matrix using all input samples. In the logical conditions, these range are used rather than some absolute threshold to make the model more dynamic and flexible to disabled persons. The fundamental control hand gestures and the underlying idea behind the development of the suggested mathematical model for gesture recognition is presented in Fig 4 which provides a visual representation of the basic control gestures and the key principles utilized in the creation of the mathematical model.

When the aforementioned 'Drive' gesture is detected for the first moment during execution, the fingertip's location is saved as the starting position. Subsequently, the index fingertip is continuously monitored to recognize the four directional operations which are: Drive left, drive right, drive forward, and drive backward, compared to the starting drive location of hand. To perform a movement in a specific direction, a certain distance from the starting position needs to be stretched by the index fingertip (keypoint 8) in the particular direction. These threshold distances are determined based on the size of the image frame which covers the hand. For instance, while moving the index finger upward, the output will be recognized as "Drive forward" if the fingertip (keypoint 8) crosses a certain distance threshold along the vertical direction while keeping a reasonable tolerance distance horizontally. This tolerance threshold is introduced to increase user flexibility. Similarly, the recognition of 'backward', 'left', and 'right' gestures will be based on the crossing of similar distance thresholds by the fingertip in their respective directions.

The logical conditions for recognizing gestures for 'Drive' in four directions are outlined in Table 3. Here, '$x_{cur}$' and '$y_{cur}$' represents the current x and y-coordinate of the tip of the index finger. And, '$x_{init}$' and '$y_{init}$' represents the initial x-coordinate and y-coordinate of the tip of the index finger when 'Drive' gesture is recognized for the first time. Again, '$mov_x$' and '$mov_y$' denotes the certain distances which index finger needs to go from initial position in x and y direction to recognize the horizontal (Left, Right) and vertical (Forward, backward) movement operations respectively. Lastly, '$tol$' represents the tolerance threshold which can be endured horizontally during the vertical movements or vertically during the horizontal movements of the index finger. It is introduced to give the user flexibility under acceptable limit to avoid frequent wrong control operation.

Fig 5 presents some sample snapshots of the four directional movement gestures taken by the RGB camera. The red dot is used to represent the initial position ($x_{init}$, $y_{init}$) of the tip of the index finger when 'Drive' gesture is detected for the first time. Then, based on the movement

Table 3. Conditions used for directional gestures in the proposed technique.

| Gestures | Parameters | Required conditions to meet |
|---|---|---|
| Drive Right | $x_{cur}, x_{init}$ | $\left(x_{cur} - x_{init}\right) > mov_x$ |
| | $y_{cur}, y_{init}$ | $tol < \left(y_{cur} - y_{init}\right) < tol$ |
| Drive Left | $x_{cur}, x_{init}$ | $\left(x_{cur} - x_{init}\right) < -mov_x$ |
| | $y_{cur}, y_{init}$ | $tol < \left(y_{cur} - y_{init}\right) < tol$ |
| Drive Forward | $y_{cur}, y_{init}$ | $\left(y_{cur} - y_{init}\right) > mov_y$ |
| | $x_{cur}, x_{init}$ | $tol < \left(x_{cur} - x_{init}\right) < tol$ |
| Drive_Backward | $y_{cur}, y_{init}$ | $\left(y_{cur} - y_{init}\right) < -mov_y$ |
| | $x_{cur}, x_{init}$ | $tol < \left(x_{cur} - x_{init}\right) < tol$ |

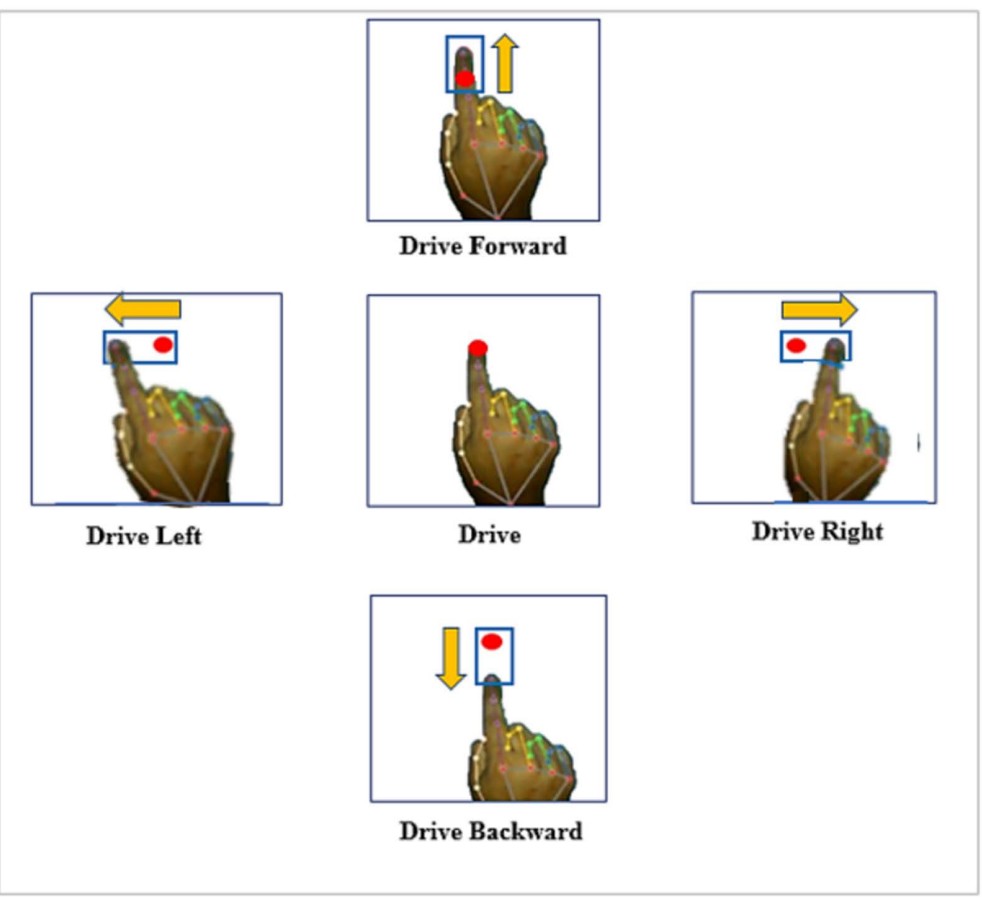

**Fig 5. Sample Snapshots of captured images of gesture recognition for directional movements.**

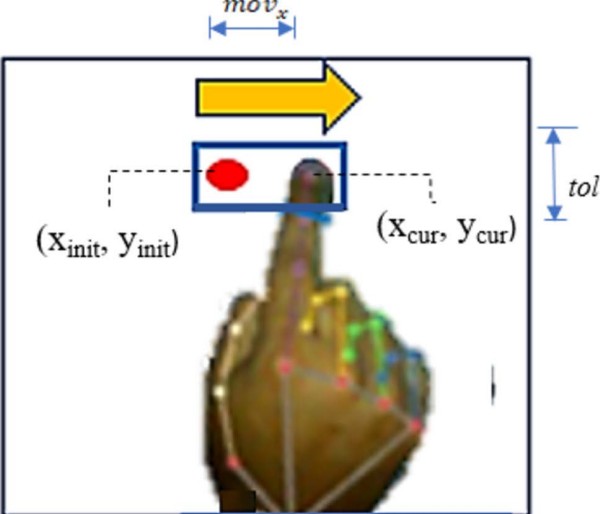

**Fig 6. Sample Snapshot of 'Drive Right' gesture and relevant parameters with thresholds.**

of the index finger from this initial position, four directional gestures are recognized upon satisfying the required logical conditions, as shown in Table 3. The user does not need to move full hand for the directional operations, rather only movement of index fingertip is enough to recognize the gestures correctly. For example, to perform the 'Drive right' gesture, if the user moves his or her index finger only in right direction, keeping the wrist in same location, still the gesture will be detected as 'Drive right' as required conditions will be met. This detection is continuous and if the user moves his finger from a direction to another, the detected gesture type will also be changed accordingly.

From the sample 'Drive Right' gesture snapshot as shown in Fig 6, it can be seen that the amount of movement of index fingertip from $(x_{init}, y_{init})$ to $(x_{cur}, y_{cur})$ is such that it crosses the threshold $mov_x$ to the right direction, while maintaining the fingertip position within a certain acceptable distance *(tol)* along the vertical direction. As per the proposed algorithm, this gesture is recognized as 'Drive Right' gesture. Similarly, other directional gestures are recognized with different logical conditions. The user needs to move his or her finger only in desired direction. It is not mandatory to move the whole hand in any direction.

### Ethics statement

Ethical approval for this experiment for testing the proposed technique by human participant, has been obtained from Institutional Review Board (IRB), BUET, Dhaka, Bangladesh (Ref/ BUET/RISE/IRB/2025-01).

## 4. Results and discussions

This part presents an analysis of the overall performance and results attained from the proposed mathematical hand gesture recognition technique designed for a smart wheelchair including a comparison with other existing methods.

### 4.1. Experimental setup

For the testing of the proposed hand gesture recognition technique, the hand was kept on a small structure having different colors in background and the RGB camera is set on top of it in such a way that it captures image of hand from top position. The camera is connected to a laptop of model HP ProBook 440 G6, through a USB cable and the program of gesture recognition algorithm is run in that laptop using Python 3.10. The whole technique was tested both in indoor and outdoor environment under daylight with a good number of hand samples captured by the camera.

The proposed technique in this work has also been tested in a real powered wheelchair system [25]. The system consists of two processing units: a Raspberry Pi 4 model B (1 GB RAM and 1.5 GHz of CPU frequency), and an Arduino Uno microcontroller. The primary processing device is Raspberry Pi. Its raw input voltage is 5V with 3A. For powering up this unit, a DC-DC buck converter is used to convert the 12V from the supply battery to 5.4V with a maximum 3A output current, which is needed for the Raspberry Pi in overclocking mode. Thus, CPU speed goes up to 2.0GHz and provides faster real-time processing. It generated the control commands by performing gesture recognition. In this system, two motor drivers and the Raspberry Pi are connected via Arduino to enable motor navigation. Through serial communication, Arduino receives instructions based on input gestures from the main processor unit (Raspberry Pi). Then, it directs particular PWM impulses at motor drivers. The system has also two DC motors having a power rating of 250W.

The control circuit of the used smart wheelchair system consists of two motor-drivers combined with an Arduino Uno CPU and their connection to a DC power source. The

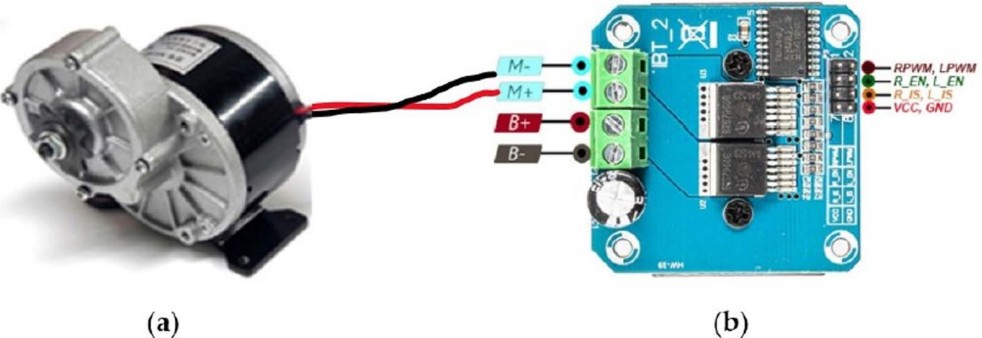

**Fig 7. Interfacing of the (a) motor and (b) moto driver used in the smart wheelchair system.**

system utilizes the MY1016Z2 motor as Fig 7 illustrates. Its power rating is 250 Watts (0.33 Horsepower), and it runs at 24 V and 13.4 Amps. This motor can rotate in both clockwise and counterclockwise directions. Its rated load RPM is 400, and its no-load RPM is 3300. The motor driver model used here is the BTS7960. This driver's specifications are appropriate and meet the requirements. Compared to the conventional four relay control circuit, this motor driver offers a more convenient H-bridge control circuit. Each motor driver has 8 control pins including + 5 V Vcc and GND; three pins of them are R_EN, RPWM, R_IS that are engaged in forwarding rotation, and the other three L_EN, LPWM, L_IS are engaged in the backward rotation. These 8 control pins are receiving signals from connected Arduino-Uno. Arduino is linked with Raspberry pi for receiving signals based on different hand gestures. Therefore,

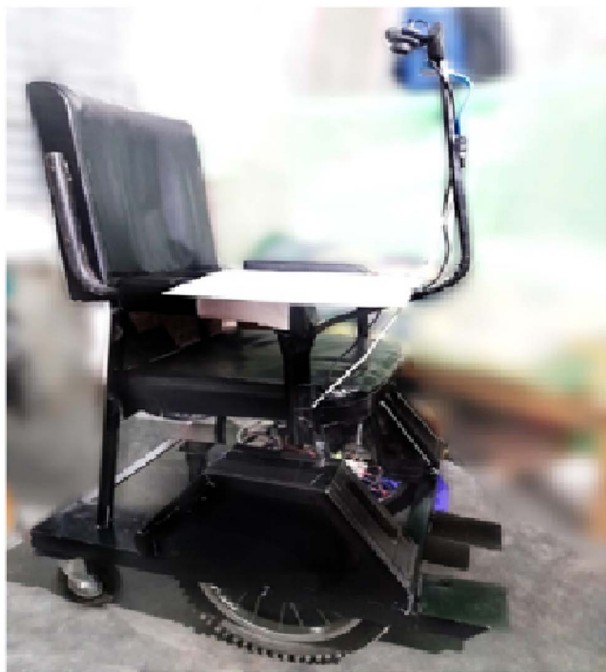

**Fig 8. The structure of the smart wheelchair system used for testing the proposed gesture recognition technique.**

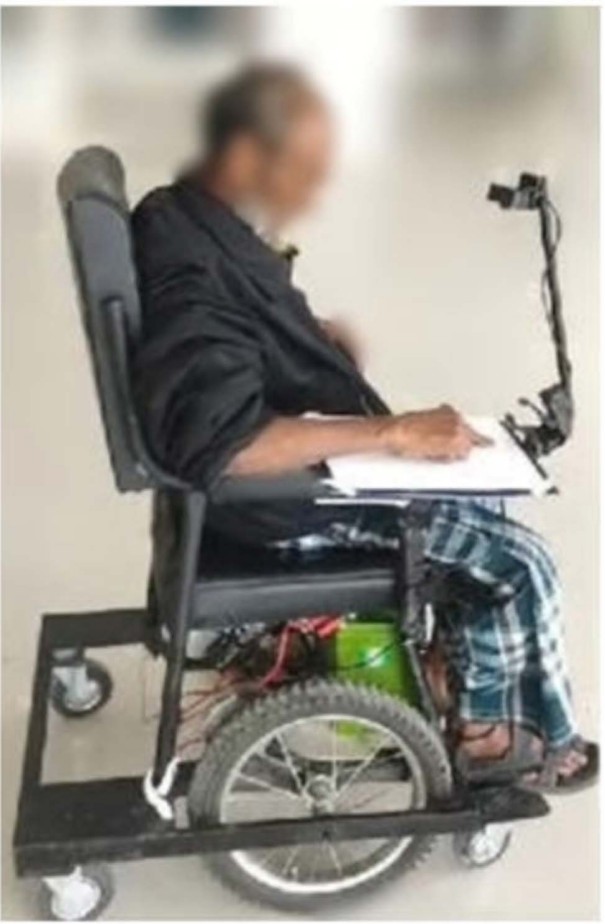

**Fig 9. Actual user operating the hand gesture-controlled wheelchair.**

circumventing specific PWM values through Raspberry pi, the motors can be operated. The complete structure of the used smart wheelchair system is shown in Fig 8. The proposed gesture recognition model has been integrated in this wheelchair system for the purpose of testing. The test was done with an actual user, whose one leg is disabled due to typhoid. A picture with the actual user is shown in Fig 9. The wheelchair performs different control operations according to different gestures by the user. We found the wheelchair is functioning properly with the proposed gesture detection mathematical model and the test has been successful.

## 4.2. Results

The proposed hand gesture recognition technique provides the gesture names as output according to the fulfillment of logical conditions of distances with thresholds. In the testing, after detecting hand for the first time under camera, it captures image for determining the three thresholds sequentially. It will ask to place hands and then calculate three thresholds Thld1, Thld2, Thld3 automatically as per the proposed algorithm. Then the program goes to the main gesture recognition module for new gesture recognition which will eventually help to perform wheelchair operations. The RGB camera captures hand image from the top of hand continuously, processes the image and detect gesture as per logical conditions which are

**Table 4. Confusion matrix of the testing of proposed hand gesture recognition technique.**

| Performed \ Detected | Stop | Horn | Drive Stop | Drive Forward | Drive Backward | Drive Left | Drive Right | Recall | Precision | F1 Score | Accuracy | Overall Accuracy |
|---|---|---|---|---|---|---|---|---|---|---|---|---|
| Stop | 98 | | 1 | | | 1 | | 0.980 | 1.000 | 0.990 | 98% | 98.14% |
| Horn | | 98 | 1 | | | | 1 | 0.980 | 1.000 | 0.990 | 98% | |
| Drive Stop | | | 99 | 1 | | | | 0.990 | 0.900 | 0.947 | 99% | |
| Drive Forward | | | 2 | 98 | | | | 0.980 | 0.990 | 0.985 | 98% | |
| Drive Backward | | | 1 | | 99 | | | 0.990 | 1.000 | 0.995 | 99% | |
| Drive Left | | | 3 | | | 97 | | 0.970 | 0.990 | 0.980 | 97% | |
| Drive Right | | | 2 | | | | 98 | 0.980 | 0.990 | 0.985 | 98% | |

Here the used gestures perform wheelchair operations as below:

**Stop:** The wheelchair will be stopped immediately from moving

**Horn:** The wheelchair will give horn

**Drive Stop:** The wheelchair will go to driving mode (standstill and ready for driving)

**Drive Forward:** The wheelchair will start moving in straight forward direction

**Drive Backward:** The wheelchair will start moving in its backward direction

**Drive Left:** The wheelchair will start moving in its left direction

**Drive Right:** The wheelchair will start moving in its right direction

established in this thesis. The output shows in real-time. If the user changes the hand gesture, the output will change accordingly.

The three basic gestures and four directional movements are taken into consideration for performance analysis. Table 4 displays the confusion matrix, where each row represents the real hand gesture and each column represents the predicted gesture. The proposed technique achieved a testing accuracy of 98.14% and a recognition rate of over 21 FPS (frames per second). The testing involved 100 samples for each hand gestures under different lightning conditions. Various performance measuring parameters are determined from the performed and predicted gestures' count. These parameters are - Precision, Recall and F1 score of the technique. The F1 score, which measures the model's ability to balance recall (capturing positive cases) and precision (accuracy of captured cases), is calculated for each gesture. The F1 score is the harmonic mean of precision and recall. Here, Precision, Recall and F1 Score is calculated using below formula.

$$Recall = \frac{TP}{TP + FN}$$

$$Precision = \frac{TP}{TP + FP}$$

$$F1Score = \frac{2 \times Recall \times Precision}{Recall + Precision}$$

$$Accuracy = \frac{TN + TP}{TN + FP + TP + FN}$$

Here,

TP = True Positive = An instance for which both predicted and actual values are positive (the technique predicts a gesture when a gesture is actually performed).

FP = False Positive = An instance for which predicted value is positive but actual value is negative (the technique predicts a gesture but no gesture is actually performed).

FN = False Negative = An instance for which predicted value is negative but actual value is positive (the technique doesn't predict a gesture but a gesture is actually performed).

TN = True Negative = An instance for which both predicted and actual values are negative (the technique doesn't predict gesture and actually no gesture is performed)

TP+FP = Total number of detected gestures by the technique for each gesture category

TP+FN = Total number of actually performed gestures for each gesture category = 100

F1 Score is calculated from Precision and Recall.

Overall Accuracy = Overall accuracy of the technique for all gestures

= (Accurately detected count/ Total sample count) × 100%

Higher precision of any algorithm means that the algorithm returns more relevant results than the irrelevant ones. It measures the percentage of predictions made by the model that are correct. High recall means that an algorithm returns most of the relevant results (whether or not irrelevant ones are also returned). It measures the percentage of relevant data points that were correctly identified by the model. F1 score is a machine learning evaluation metric that measures a model's accuracy. It combines the precision and recall scores of a model. Notably, the proposed technique achieved an F1 score greater than 0.9 for all seven gestures.

It can be seen that few gestures are recognized incorrectly during each type of gesture recognition. To check the accuracy of each gesture separately, continuous 100 samples were taken for each gesture using different sizes of hands of multiple people of different ages in varying lighting conditions. For some cases, the fingers were not be placed properly as per defined gestures. For example, during the incorrect Horn gesture, the index finger slightly came out which falls in the range of Threshold1. Similarly, for the wrongly recognized gestures, most are recognized as 'Drive Stop' gesture. This might happen due to the slight crossing of thresholds' minimum value or the size of index finger for different persons, leading to incorrect detection. Further precise logical conditions can be applied to increase the accuracy.

Also, an evaluation synopsis of associated researches is presented in Table 5. Amid these works, Gao et al. [19] proposed a recognition system that necessitates lifting a hand in order to perform any gesture, but it's ineffectual for individuals having mobility impairment and

**Table 5. Comparison of related work with the proposed technique.**

| Features | Proposed Techniques | | | |
|---|---|---|---|---|
| | Gao et al [19]. | Oliver et al [21]. | Sadi et al [25]. | Proposed Technique |
| Functionality | Hand gesture recognition | Hand movement detection | Hand gesture recognition | Hand gesture recognition |
| Requirements and Limitations | Require hand raising, background complexity | Requires wearing hand band | Requires finger movement, RGB Camera, Raspberry Pi. It doesn't work well under daylight or if the background and skin tone matches. | Requires finger movement, RGB Camera. A mathematical model is utilized. Performs well at indoor or outdoor, irrespective of background color. |
| Major Equipment | Microsoft Kinect Camera, highly configured laptop | Accelerometer, Joystick Manipulator | RGB Camera, Raspberry Pi | RGB Camera (for gesture recognition) |
| Cost | High | Medium | Low | Low |
| Recognition Success rate | 10-100% (Depends on background complexity) | Actual accuracy is not measured | 97.14% | 98.14% |

unintuitive. Also, it uses a Microsoft Kinect Depth camera which is expensive and the performance severely degrades when background complexity arises. Oliver et al.'s [21] method of using an accelerometer, which requires movement to function and needs hand band wearing, is not effective for many people with dexterity issues. Our suggested model solves these problems by allowing users to safely steer using just their fingers' movement. The users will be able to operate it easily with less energy. A system suggested by Sadi et al. [25] using 2D CNN and skin segmentation also works by finger movement, but it creates problem if the background color matches with the skin tone in any way or under daylight. By utilizing a mathematical model, our proposed technique also solved these issues. This model functions well on any background or in daylight and for any size of hands in any environment. This made the model more resilient and efficient. So, this proposed system will be very useful for the disabled people by giving them highest user friendliness with highest accuracy. Moreover, the proposed technique achieved an F1 score greater than 0.9 for all seven gestures, indicating that it predicts each observation with a high level of relevancy and accuracy.

There is an improvement in this work compared to our previous work [27] on the same topic. In our previous work, all the thresholds were considered as fixed threshold values rather than a range of maximum and minimum threshold values. Moreover, those thresholds were not automatically calculated from hand samples. That is why the technique was less dynamic and sometimes the output was misleading. Again, the technique was not working good for some special disability cases. For example, for the 'Drive' gesture, detection was not possible for those users who cannot make his index finger fully straight for any illness or disabilities. Hence, the distance between wrist to fingertip didn't meet the condition with threshold. Keeping this in mind, a maximum and minimum values were determined for all thresholds at the beginning of hand detection for once. After using this range concept, it would be easy to meet the thresholds and making the technique more dynamic and more useful for disabled people. Fig 10 to Fig 13 illustrates these issues and comparison between our old and new methodology.

Fig 10 and Fig 11 illustrate the comparison between response of our previous and current proposed methodology for 'Drive' gesture for the case of a disabled person who cannot make his index finger straight properly. In our old methodology, the threshold Thld1 is considered

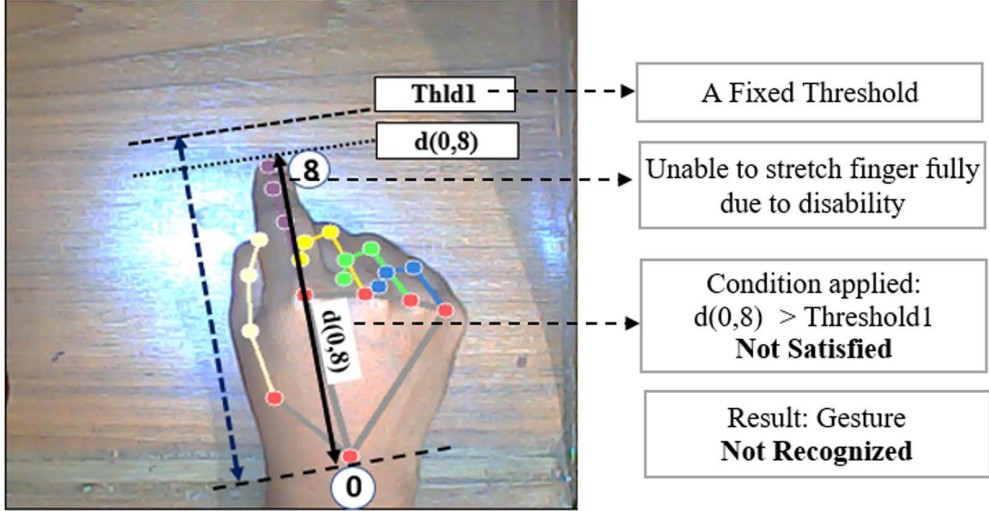

**Fig 10. Problem in 'Drive' gesture recognition in our previous methodology** [27].

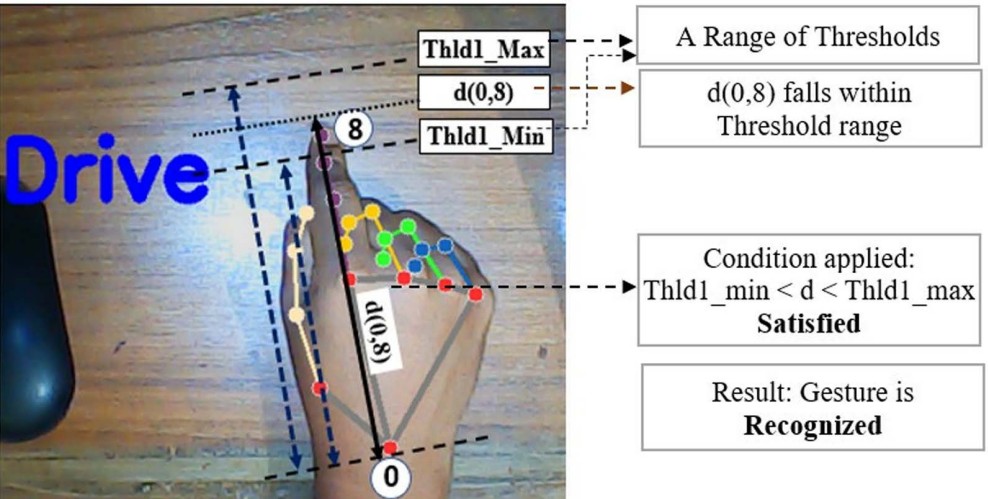

**Fig 11. Successful 'Drive' gesture recognition in current proposed methodology.**

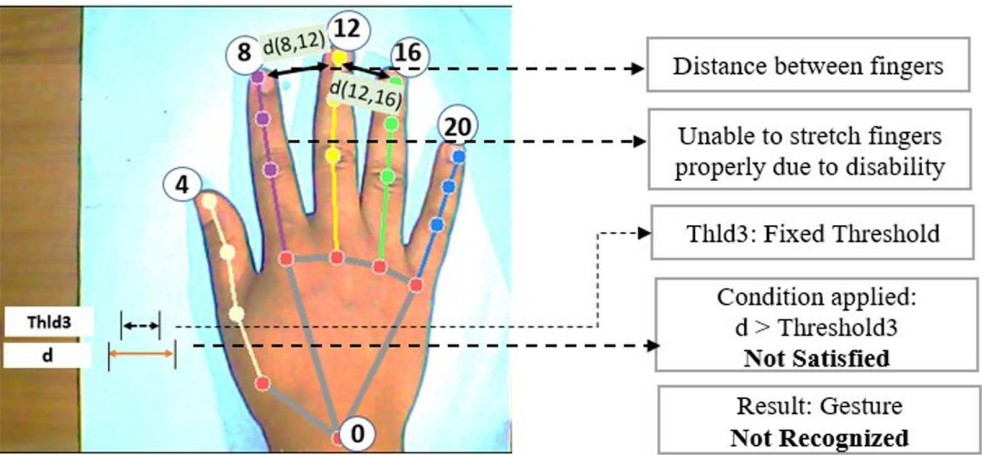

**Fig 12. Problem in 'Stop' gesture recognition in our previous methodology [27].**

a fixed threshold always. As a result, the technique failed to detect the gesture for that disabled person because d(0,8) couldn't cross the threshold Thld1. But, in the new proposed methodology, a maximum and minimum range of Thld1 is used rather than fixed threshold and distance d(0,8) falls between the maximum and minimum range of Thld1 and therefore the gesture is recognized properly. This makes the proposed technique more flexible and efficient. The threshold range is determined from distance matrix of live samples of hand so that the range doesn't become too high or too small which could increase false detection probability.

Fig 12 and Fig 13 illustrate the comparison between response of our previous and current proposed methodology for 'Stop' gesture for the case of a disabled person who are unable to stretch his fingers properly. Here, due to applying range of threshold for Thld3 in proposed

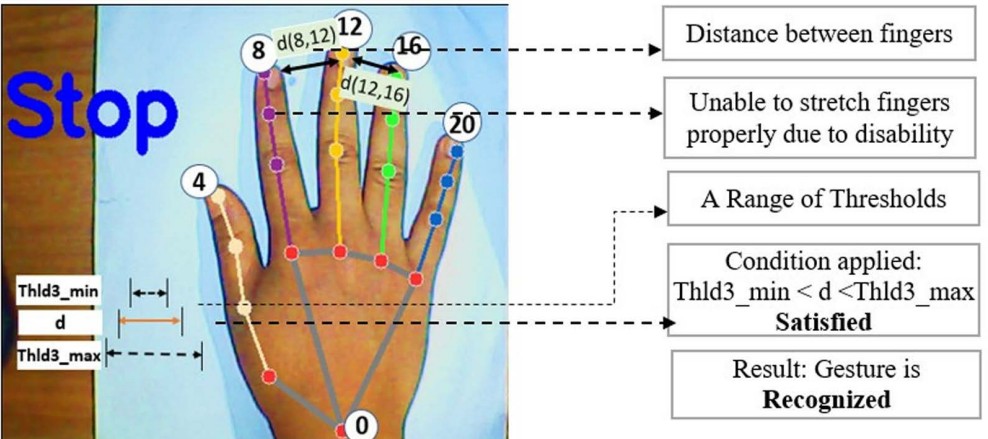

**Fig 13. Successful 'Stop' gesture recognition in current proposed methodology.**

**Table 6. Confusion matrix for previous methodology of hand gesture recognition technique [27].**

| Performed \ Prediction | Stop | Horn | Drive Stop | Drive Forward | Drive Backward | Drive Left | Drive Right | Recall | Precision | F1 Score | Accuracy | Overall Accuracy |
|---|---|---|---|---|---|---|---|---|---|---|---|---|
| Stop | 97 | 1 | 2 | | | | | 0.970 | 1.000 | 0.985 | 97% | 97.00% |
| Horn | | 98 | 2 | | | | | 0.980 | 0.951 | 0.966 | 98% | |
| Drive Stop | | 4 | 95 | 1 | | | | 0.950 | 0.888 | 0.918 | 95% | |
| Drive Forward | | | 2 | 97 | | | 1 | 0.970 | 0.970 | 0.970 | 97% | |
| Drive Backward | | | 2 | | 98 | | | 0.980 | 1.000 | 0.990 | 98% | |
| Drive Left | | | 3 | 1 | | 96 | | 0.960 | 1.000 | 0.980 | 96% | |
| Drive Right | | | 1 | 1 | | | 98 | 0.980 | 0.990 | 0.985 | 98% | |

new method, rather than a fixed threshold value like old method, the technique enables correct recognition of 'Stop' gesture. In the new proposed technique, if the distance between the fingertips are maintained such that it remains inside the threshold range, then the gesture will be detected. But in old technique, it always required to cross a fixed threshold for 'Stop' gesture detection which might be difficult for some disabled aged persons. Thus, the proposed new technique becomes more suitable, dynamic and flexible and will work for different types of disabled and aged persons who will use the wheelchair.

Table 6 displays the confusion matrix using our previous methodology [27] for similar number of input samples used in proposed new methodology. It can be seen that, the average F1 score was 0.97 and the Accuracy was 97% which is poorer than our current methodology (Table 4). Specially the performance issues due to considering fixed threshold values rather than a range, which are demonstrated in Fig 10 to Fig 13, is also reflected in the confusion matrix. So, our current methodology performs better than the old one after considering maximum and minimum range of thresholds instead of fixed thresholds. It will make the proposed technique more usable and flexible to the users having different types of dexterity issues.

## 5. Conclusions

The primary objective of this work is the development of a real-time hand gesture recognition technique that enables controlling wheelchair in the most flexible way using finger movements of hands of any size, in both indoor and outdoor lightening condition. In the existing methodologies of various researchers, their proposed techniques are not so suitable and efficient for users having dexterity problems and some techniques developed using skin segmentation and CNN, don't work well under daylight on any background. In light of these difficulties, a novel mathematical model has been designed using the positions and distances among major hand landmarks and a few thresholds. The basic three wheelchair operations- 'Drive', 'Stop', 'Horn', along with driving in four directions are implemented. The proposed model has also been experimented in an electric wheelchair. The experimental results show that the technique suggested in this work achieved a testing accuracy of 98.14%, with a recognition speed of greater than 21 FPS, whereas other related methodologies obtained variable success rate between 10–100% and the recent methodology achieved 97.14%. So, the proposed methodology outperforms the existing methodologies. Our proposed technique has the capability to control a wheelchair effectively in a real-time environment using the movement of fingers only. So, it's more flexible for disabled and elderly people than other existing systems. It works well under daylight or in indoor environment. We hope to extend more advanced features in the future for implementing more useful control operations of wheelchair.

## 6. Suggestion for future work

In the proposed technique, basic wheelchair operations have been implemented by gesture recognition. In future, further useful control operations can be developed which may include- changing speed of wheelchair, fall detection, emergency brake, etc. Incorporating these functionalities will give the users more security and reliability. These may require recognition of more different gestures other than used gestures in this paper, where new conditions on hand landmarks with thresholds and critical calculations might be required. Also, an online monitoring system or android application can be developed to monitor wheelchair users' real-time status to ensure safety and analyze their daily activities from the data which may be used for further research and for the treatment of the disabled people. An alarm system can be developed in case of any fall detection or long-time inactivity of the wheelchair users. Moreover, more precision calculation on thresholds range can be explored to achieve higher recognition success rate. A qualitative study on higher number of individuals with disabilities and testing in a specific route with blockages can be conducted with more evaluation matrices.

## Acknowledgments

Institute of Information and Communication (IICT), Bangladesh University of Engineering and Technology (BUET), is to be expressed gratitude on behalf of the authors for providing all necessary assistance to conduct this study.

## Author contributions

**Conceptualization:** Md. Liakot Ali, Muhammad Sheikh Sadi.

**Data curation:** Md Rafiul Huda.

**Formal analysis:** Md Rafiul Huda.

**Investigation:** Md. Liakot Ali, Md Rafiul Huda.

**Methodology:** Md. Liakot Ali, Md Rafiul Huda, Muhammad Sheikh Sadi.

**Resources:** Muhammad Sheikh Sadi.

**Software:** Md Rafiul Huda.

**Supervision:** Md. Liakot Ali, Muhammad Sheikh Sadi.

**Validation:** Md. Liakot Ali, Md Rafiul Huda.

**Visualization:** Md. Liakot Ali.

**Writing – original draft:** Md Rafiul Huda.

**Writing – review & editing:** Md. Liakot Ali, Md Rafiul Huda, Muhammad Sheikh Sadi.

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
