## [Decision Letter · Decision Letter 0]

22 May 2024

Dear Dr. Ali,

Thank you for submitting your manuscript to PLOS ONE. After careful consideration, we feel that it has merit but does not fully meet PLOS ONE’s publication criteria as it currently stands. Therefore, we invite you to submit a revised version of the manuscript that addresses the points raised during the review process.

Thank you for your submission to PLOS One. On the positive side, the reviewers positively evaluate the development of the proposed gesture recognition system for a smart wheelchair. They raise some points that should be addressed prior to publication, such as clearly stating testing with disabled users as a limitation and future direction, given that testing reported in the manuscript appears to have been conducted with non-disabled users.

On the negative side, R1 points out notable similarities between the current manuscript and this one (https://ieeexplore.ieee.org/stamp/stamp.jsp?arnumber=10088702&casa_token=hy9pM5Prw5YAAAAA:Bqlb_FgBKzwaJt5GCTtd2pRP_p9BAOikYZrxLtIFL_aNJuwA0kZRUVFG9wnTpyoxv9FuhSU&tag=1). After having compared these two manuscripts myself, a stronger justification is warranted for how they differ. Without such a justification, the manuscript will be rejected, per PLOS One policy.

Due to an undisclosed COI between R1 and one of the authors, if this manuscript is sent out for an additional round of review, R1 will not be asked to re-review the manuscript and a review from an additional reviewer will be solicited.

We look forward to receiving your revised manuscript.

Kind regards,

Laura Morett

Academic Editor

PLOS ONE

Journal Requirements:

3. We note that your Data Availability Statement is currently as follows: [All relevant data are within the manuscript and its Supporting Information files]

Reviewers' comments:

Reviewer's Responses to Questions

**Comments to the Author**

1. Is the manuscript technically sound, and do the data support the conclusions?

Reviewer #1: Yes

Reviewer #2: Yes

2. Has the statistical analysis been performed appropriately and rigorously?

Reviewer #1: Yes

Reviewer #2: Yes

3. Have the authors made all data underlying the findings in their manuscript fully available?

Reviewer #1: Yes

Reviewer #2: Yes

4. Is the manuscript presented in an intelligible fashion and written in standard English?

Reviewer #1: Yes

Reviewer #2: Yes

Reviewer #1: This paper presents a new mathematical model for an efficient and real-time hand gesture recognition technique for an intelligent wheelchair control system, which aligns with contemporary technologies. The model is developed using the positions of the significant hand landmarks, distances among them and some critical thresholds, which are determined by a good number of samples of hands. The proposed method offers more flexibility for users through less hand movement when operating the wheelchair. The model is tested for hands of different sizes, irrespective of any background indoors or outdoors under sunlight. The experimental study demonstrates that it outperforms the existing methods regarding success rate with good performance metrics.

This paper is well-written and organized, with sufficient figures and tables describing the methodology behind the architecture and supporting the experimental results. In summary, this paper innovates in real-time hand-gesture recognition techniques for wheelchair control. It also performs extensive experiments and statistical analysis to support its claims. This paper would be a valuable addition to the reputed PLoS ONE journal.

However, some minor issues need to be focused on before final acceptance.

• Please add your contributions in bullet points for clarity in the Introduction section.

• It is better to use the same convention to present the same thing. For example, “Tejonidhi et al. [24]” and “Sadi, M.S. et al. [25]” should follow the same presentation style.

• Please provide a summary table for the works mentioned in the Related Work section. This addition would give an in-brief overview of the existing state-of-the-art.

• I noticed that both “Experimental Setup” and “Results” started with the numbering of 4.1. Are they correct?

• The work presented in [27] obtained an accuracy of 99.17%, whereas this work achieved an accuracy of 98.14%. Also, many results are presented in this work from [27]. Please explain in detail the superiority of this work over [27].

• I did not find any differences between a few of the figures presented in [27] and this work. The figures are Fig. 1, Fig. 2, Fig. 4 and Fig. 5 [27], Fig. [5], and Fig. 6 [27]. Could you please clarify these things?

• Please use the same reference style for all references. For example, [15], [16], [21]-[23], and [25]-[26] have been deviated from other reference style. It is better to check all the references again to add the missing volume, page number, issue, etc.

• Please add a picture of the actual users, operating the wheelchair.

Reviewer #2: This paper presents a new mathematical model for an efficient and real-time hand gesture recognition technique for an intelligent wheelchair control system. The paper reviews current techniques, highlights the advantages of the proposed method, and presents experimental results obtained by applying the method. The experimental study demonstrates that it outperforms existing methods in terms of success rate with good performance values. However, I consider that the authors need to clarify the following aspects:

1. The method presented is dedicated to individuals with major motor disabilities who cannot properly use a joystick. In the vast majority, these individuals cannot properly use their fingers either. The study appears to be conducted on a sample of able-bodied individuals. A qualitative study conducted on individuals with disabilities is lacking.

2. Additionally, the method presented eliminates the joystick as a means of control. However, the joystick also controls the speed of movement. There is a reference in the paper to further development of speed control. I believe this should be expanded upon, presenting at least a summary of the algorithm for gradual speed increase/decrease, as well as justification for choosing the optimal speed for long distances.

3. The control part of the wheelchair motors is not described in terms of method and control parameters, especially in the implementation on a real wheelchair.

4. Forward/backward/lateral movements by pointing with the index finger work for able-bodied individuals, but what is the error rate relative to the level of disability?

5. It might have been beneficial to include a medical perspective, outlining the types and degrees of motor disabilities that can access this technique, as well as the method's advantages. Additionally, it could identify possible and potentially easier types of gestures.

6. Although a known gesture recognition method is mentioned, it may be helpful to mention at least the main stages of the method.

**Do you want your identity to be public for this peer review?** For information about this choice, including consent withdrawal, please see our Privacy Policy

Reviewer #1: No

Reviewer #2: **Yes: ** Ionel Cristian Vladu

---

## [Author Response · Author response to Decision Letter 1]

24 Sep 2024

All respected reviewers' comments have been addressed and responses have been provided inside the 'Response to Reviewers Comments' file accordingly.

RESPONSES TO REVIEWERS’ COMMENTS :

We would like to thank the reviewers for their careful read and constructive feedback on the previous draft entitled “Developing a Real-Time Hand-Gesture Recognition Technique for Wheelchair Control”. We have carefully considered their comments in preparing our revision, which has resulted in a manuscript that is clearer, more compelling and broader.

Reviewers' Comments:

....................................................................................................................................................................

Reviewer #1:

Comment #1: Please add your contributions in bullet points for clarity in the Introduction section.

Response: Thank you for your humble consideration and recommendation. The contributions of our paper have been added in bullet points in the Introduction section. [Please see Introduction (Section 1) in page 4 of the revised manuscript].

Contributions of this research are -

• A unique mathematical model has been designed utilizing hand landmarks and some thresholds for hand gesture recognition

• An efficient, flexible gesture recognition system for the control of a wheelchair has been proposed which requires only movement of fingers

• A better gesture recognition system for wheelchair control has been proposed focusing these aspects all together: users’ flexibility, high accuracy, and environment (indoor and outdoor) independency.

Comment #2: It is better to use the same convention to present the same thing. For example, “Tejonidhi et al. [24]” and “Sadi, M.S. et al. [25]” should follow the same presentation style.

Response: Thank you for your thoughtful observation. The mentioned presentation style has been corrected in revised manuscript and same convention has been used to present the same things. [Please see Related Work (Section 2) in page 6 of the revised manuscript].

Sadi et al. [25] designed a wheelchair system which is controlled by hand gestures in real time, where the authors used YCrCb skin segmentation technique, Haar-cascade classifier, KCF tracking algorithm, and a 2D CNN model.

Comment #3: Please provide a summary table for the works mentioned in the Related Work section. This addition would give an in-brief overview of the existing state-of-the-art.

Response: Thank you for your insightful comment and recommendation. A summary table with overview of related works has been added in revised manuscript. [Please see Table 1 under Related Work (Section 2) in page 7 of the revised manuscript].

Comment #4: I noticed that both “Experimental Setup” and “Results” started with the numbering of 4.1. Are they correct?

Response: Thank you for your precise observation. Sorry for the inconvenience. The Section numbering of “Results” section has been corrected in revised manuscript. [Please see “Results” in page 18 of the revised manuscript].

Comment #5: The work presented in [27] obtained an accuracy of 99.17%, whereas this work achieved an accuracy of 98.14%. Also, many results are presented in this work from [27]. Please explain in detail the superiority of this work over [27].

Response: Thank you for your incisive statement and observation. The methodology used in the work presented in [27] is different from the proposed new methodolgy in this paper. In the previous work of [27], all the thresholds were considered as fixed values but in the new proposed methodology, a range of maximum and minimum threshold values have been used which has made the system more dynamic, accessible and it showed better performance. Though the accuracy in the work presented in [27] was higher, but sample count was very low. In the work [27], during performance evaluation, only 20 samples were taken for each gesture, which was not reflecting the actual efficiency of the system as it considered lower variety of sample. Whereas, in the proposed new methodology, 100 samples were used for each gesture for evaluating performance which gives better reflection. For fair comparison, the old methodology was also tested for similar number of samples (100 per gesture) and found accuracy is lower (97%), whereas, the proposed new methodology attained higher accuracy (98.14%). So, comparing fairly, the proposed new methodology is performing better than the previous one. The confusion matrix for old methodology of [27] for similar number of samples has been provided in the manuscript. [Please see page 22-25 of the revised manuscript (under section 4.2)].

There is an improvement in this work compared to our previous work [27] on the same topic. In our previous work, all the thresholds were considered as fixed threshold values rather than a range of maximum and minimum threshold values. Moreover, those thresholds were not automatically calculated from hand samples. That is why the technique was less dynamic and sometimes the output was misleading. Again, the technique was not working good for some special disability cases. For example, for the ‘Drive’ gesture, detection was not possible for those users who cannot make his index finger fully straight for any illness or disabilities. Hence, the distance between wrist to fingertip didn’t meet the condition with threshold. Keeping this in mind, a maximum and minimum values were determined for all thresholds at the beginning of hand detection for once. After using this range concept, it would be easy to meet the thresholds and making the technique more dynamic and more useful for disabled people. Fig. 8 to Fig. 11 illustrates these issues and comparison between our old and new methodology.

Comment #6: I did not find any differences between a few of the figures presented in [27] and this work. The figures are Fig. 1, Fig. 2, Fig. 4 and Fig. 5 [27], Fig. [5], and Fig. 6 [27]. Could you please clarify these things? .

Response: Thank you for your intuitive observation. Fig. 1 of the manuscript contains the major steps of the hand gesture recognition technique which is same for [27] but the methodology, thresholds calculation and samples count is different in these two works. Fig. 2 represents the positions and names of the major hand landmarks in a hand which is universally used for the framework used for hand gesture recognition in the two works which is used for explaining the steps. Fig. 4, Fig. 5 [27], Fig. [5] and Fig. 6 [27] contains the classes of hand gestures used in the proposed technique for performing different operations of wheelchair (Stop, Horn, Drive). Same gestures has been used in the proposed new methodology for fair comparison and performance evaluation.

Comment #7: Please use the same reference style for all references. For example, [15], [16], [21]-[23], and [25]-[26] have been deviated from other reference style. It is better to check all the references again to add the missing volume, page number, issue, etc.

Response: Thank you for your important observation and comment. Sorry for the inconvenience. Reference styles have been corrected where required and same reference style have been used for all references in the revised manuscript.

Comment #7: Please add a picture of the actual users, operating the wheelchair.

Response: Thank you for your suggestion and recommendation. The picture of actual users, operating the wheelchair have been added in the revised manuscript. [Please see Fig. 9 at page 19 of the revised manuscript (under section 4.1)].

Reviewer #2:

Comment #1: The method presented is dedicated to individuals with major motor disabilities who cannot properly use a joystick. In the vast majority, these individuals cannot properly use their fingers either. The study appears to be conducted on a sample of able-bodied individuals. A qualitative study conducted on individuals with disabilities is lacking.

Response: Thank you for your insightful observation and comment. The methodology presented in the manuscript will be applicable only for those disabled persons who can move only their fingers of one hand. Operating the joystick requires more energy and muscle power than moving only fingers. The proposed methodology has also been tested by a sample individual with disabilities. [Please see Fig. 9 at page 19 of the revised manuscript]. A qualitative study on a good number of individuals with disabilities is planned to be conducted in near future.

Comment #2: Additionally, the method presented eliminates the joystick as a means of control. However, the joystick also controls the speed of movement. There is a reference in the paper to further development of speed control. I believe this should be expanded upon, presenting at least a summary of the algorithm for gradual speed increase/decrease, as well as justification for choosing the optimal speed for long distances.

Response: Thank you for your comment and recommendation. The major four wheelchair operations have been implemented in the paper. There is plan to develop more features under future works including the speed control, optimal speed selection. To implement these, we may have to use different new classes of hand gestures and establish new mathematical equations using positions and distances among the major landmarks and thresholds.

Comment #3: The control part of the wheelchair motors is not described in terms of method and control parameters, especially in the implementation on a real wheelchair.

Response: Thank you for your important remark and observation. The major four wheelchair control operations have been implemented and described in the paper. Upon satisfying the mathematical conditions, different gestures are recognized and different control operations are performed as per methodology where wheelchair motors are controlled with the support of Arduino-Uno and Raspberry pi. The description of the control circuit of the wheelchair motors has been included in revised manuscript. [Please see page 18 of the revised manuscript (under Section 4.1)].

The control circuit of the used smart wheelchair system consists of two motor-drivers combined with an Arduino Uno CPU and their connection to a DC power source. The system utilizes the MY1016Z2 motor as Fig. 7 illustrates. Its power rating is 250 Watts (0.33 Horsepower), and it runs at 24 V and 13.4 Amps. This motor can rotate in both clockwise and counterclockwise directions. Its rated load RPM is 400, and its no-load RPM is 3300. The motor driver model used here is the BTS7960. This driver's specifications are appropriate and meet the requirements. Compared to the conventional four relay control circuit, this motor driver offers a more convenient H-bridge control circuit. Each motor driver has 8 control pins including +5 V Vcc and GND; three pins of them are R_EN, RPWM, R_IS that are engaged in forwarding rotation, and the other three L_EN, LPWM, L_IS are engaged in the backward rotation. These 8 control pins are receiving signals from connected Arduino-Uno. Arduino is linked with Raspberry pi for receiving signals based on different hand gestures. Therefore, circumventing specific PWM values through Raspberry pi, the motors can be operated.

Comment #4: Forward/backward/lateral movements by pointing with the index finger work for able-bodied individuals, but what is the error rate relative to the level of disability?

Response: Thank you for your astute observation and comment. The Forward/backward/lateral movements by pointing with the index finger will work with same accuracy in similar way for the disabled persons who have at least the ability to move fingers of the hands to perform the gestures.

Comment #5: It might have been beneficial to include a medical perspective, outlining the types and degrees of motor disabilities that can access this technique, as well as the method's advantages. Additionally, it could identify possible and potentially easier types of gestures.

Response: Thank you for your insightful comment and observation. The disabled individuals who have the ability to move the fingers of the hand at least, can use the proposed technique. The technique has been tested in a real electric wheelchair operated by an actual disabled person also. A further detailed assessment and medical study with individuals with various degrees of motor disabilities couldn’t be performed due to permission issue.

Comment #6: Although a known gesture recognition method is mentioned, it may be helpful to mention at least the main stages of the method.

Response: Thank you for you observation and recommendation. The proposed gesture recognition technique has been clearly described in detailed steps in the paper. [Please see page 8 to page 16 of the revised manuscript (Section 3)]. The proposed technique utilizes a robust framework named MediaPipe Hands, for tracking the hands and fingers only which is a part of those steps. The short description about how MediaPipe Hands work, has been given in the revised manuscript also. [Please see page 9 of the revised manuscript (under Section 3.1)].

---

## [Decision Letter · Decision Letter 1]

3 Dec 2024

Dear Dr. Ali,

Thank you for submitting your manuscript to PLOS ONE. After careful consideration, we feel that it has merit but does not fully meet PLOS ONE’s publication criteria as it currently stands. Therefore, we invite you to submit a revised version of the manuscript that addresses the points raised during the review process.

**I thank the authors for their patience.  As R1 was unable to review a revision of the manuscript, an additional reviewer (R3) was recruited.  This reviewer raises a few minor concerns as well as two more major ones in which they suggest additional data collection.  Although I am not convinced that additional data collection is necessary, I ask that the authors engage with the points raised by R3 and address them in a revision.**

We look forward to receiving your revised manuscript.

Kind regards,

Laura Morett

Academic Editor

PLOS ONE

Reviewers' comments:

Reviewer's Responses to Questions

**Comments to the Author**

Reviewer #2: All comments have been addressed

Reviewer #3: (No Response)

2. Is the manuscript technically sound, and do the data support the conclusions?

Reviewer #2: Yes

Reviewer #3: Yes

3. Has the statistical analysis been performed appropriately and rigorously?

Reviewer #2: Yes

Reviewer #3: Yes

4. Have the authors made all data underlying the findings in their manuscript fully available?

Reviewer #2: Yes

Reviewer #3: Yes

5. Is the manuscript presented in an intelligible fashion and written in standard English?

Reviewer #2: Yes

Reviewer #3: Yes

**Reviewer #2: ** The identified gaps and ambiguities have been appropriately addressed. I consider that the paper can be published in this form.

**Reviewer #3: ** The paper presents a novel method of human-wheelchair interaction that uses a camera to read user's hand gesture which later translate into wheelchair motion. It is real time efficient approach. However, it is tested on 1 participant only and the design of the experiment is not clear.

1. Participants: The manuscript is missing Ethics Statement. As it involves a human participant.

2. The Table 1 have mistakes. Here are two numbers I found to be wrong. [22] and [23]' accuracies are wrong. Need revision.

3. Table 1 has column for equipment and then list MATLAP, modified VGG-8 model and ROS as equipment. Need fix because they are not equipments.

4. Figures are not places within the pdf but shown at the end. Need to be placed in the pdf.

5. Details about the 100 images collected with different size and background are missing. "The testing involved 100 samples for each hand gestures under different lightning conditions"

6. Big weakness this paper has is the small number of participants and the experiment is not clear. I understand it is difficult to hire people with difficulties. Thus, having one user from the target audience is a acceptable. However, authors can recruit able bodied people. Second point about the experiment. The author should design better experiments. Current experiment is testing the gesture while the wheelchair not moving and is designed only to test the ability of the system to read the gesture. But they did not test whether the user able to navigate within a specific route efficiently and safely. In conclusion, I suggest the authors hire more participants (not necessarily disabled) and design an experiment that test how long would it take the user to finish a navigation task and how comfortable and safe they are while navigating.

**Do you want your identity to be public for this peer review?** For information about this choice, including consent withdrawal, please see our Privacy Policy

Reviewer #2: No

Reviewer #3: No

---

## [Author Response · Author response to Decision Letter 2]

17 Jan 2025

RESPONSES TO REVIEWERS’ COMMENTS

We would like to thank the reviewers for their careful read and constructive feedback on the previous draft entitled “Developing a Real-Time Hand-Gesture Recognition Technique for Wheelchair Control”. We have carefully considered their comments in preparing our revision, which has resulted in a manuscript that is clearer, more compelling and broader.

Reviewers' Comments:

Reviewer #3:

Comment #1: Participants: The manuscript is missing Ethics Statement. As it involves a human participant.

Response: Thank you for your valuable observation. Ethics Statement: Ethical approval for this experiment for testing the proposed technique by human participant, has been obtained from Institutional Review Board (IRB), BUET, Dhaka, Bangladesh (Ref/BUET/RISE/IRB/2025-01). [Please see Section 4.1 in page 19 of the revised manuscript]. The Ethical Approval letter also uploaded with manuscript.

Ethics Statement: Ethical approval for this experiment for testing the proposed technique by human participant, has been obtained from Institutional Review Board (IRB), BUET, Dhaka, Bangladesh (Ref/BUET/RISE/IRB/2025-01).

Comment #2: The Table 1 have mistakes. Here are two numbers I found to be wrong. [22] and [23]' accuracies are wrong. Need revision.

Response: Thank you for your precise observation. Sorry for the inconvenience The accuracies of mentioned research works have been revised in Table 1 accordingly. [Please see Table 1 under Related Work (Section 2) in page 7 of the revised manuscript].

Comment #3: Table 1 has column for equipment and then list MATLAP, modified VGG-8 model and ROS as equipment. Need fix because they are not equipments.

Response: Thank you for your incisive statement and observation. The ‘Major Equipment’ column of Table 1 has been revised for relevant research works, accordingly. Only actual equipment names were kept. [Please see Table 1 under Related Work (Section 2) in page 7 of the revised manuscript].

Comment #4: Figures are not places within the pdf but shown at the end. Need to be placed in the pdf.

Response: Thank you for your important observation. As per the ‘Manuscript Organization’ section under ‘Submission Guidelines’ of PLOS One, only Figure captions were inserted immediately after the first paragraph in which the figure was cited. Figure files have been uploaded separately. The pdf was automatically built in system.

Comment #5: 5. Details about the 100 images collected with different size and background are missing. "The testing involved 100 samples for each hand gestures under different lightning conditions"

Response: Thank you for your valuable comment and observation. Actually, for testing the proposed technique, for any specific gesture recognition, 100 samples were captured, which means, that gesture was performed 100 times, and for each time, hand landmarks’ co-ordinates are extracted, comparison done between distances and thresholds, and, if the mentioned logical conditions were met, then the relevant gesture was detected and a wheelchair operation was performed. The times these samples were taken and gestures were detected were counted. The detailed extracted data (co-ordinates of hand landmarks, distance matrix, detected gesture, thresholds) of 100 samples for each gesture have been included in ‘Data_Gesture_Recognition_extraction_outputs’ file accordingly.

Comment #6: Big weakness this paper has is the small number of participants and the experiment is not clear. I understand it is difficult to hire people with difficulties. Thus, having one user from the target audience is a acceptable. However, authors can recruit able bodied people. Second point about the experiment. The author should design better experiments. Current experiment is testing the gesture while the wheelchair not moving and is designed only to test the ability of the system to read the gesture. But they did not test whether the user able to navigate within a specific route efficiently and safely. In conclusion, I suggest the authors hire more participants (not necessarily disabled) and design an experiment that test how long would it take the user to finish a navigation task and how comfortable and safe they are while navigating.

Response: Thank you for your thoughtful observation. The proposed technique has been tested by an actual disabled person after taking proper consent to check whether it’s working or not properly. However, a qualitative study on a good number of individuals with disabilities and able-bodied people has been planned to be conducted some time soon, maintaining proper processes, to validate the proposed technique more strongly. Moreover, in order to test the actual usability of the proposed technique, an experimental evaluation in a specific test area with some routes with blockages is planned to be performed in near future where different evaluation metrices such as: navigation time, number of collisions, user feedback will be evaluated for better assessment.

---

## [Decision Letter · Decision Letter 2]

12 Feb 2025

Developing a Real-Time Hand-Gesture Recognition Technique for Wheelchair Control

PONE-D-24-10717R2

Dear Dr. Ali,

We’re pleased to inform you that your manuscript has been judged scientifically suitable for publication and will be formally accepted for publication once it meets all outstanding technical requirements.

Kind regards,

Laura Morett

Academic Editor

PLOS ONE

Additional Editor Comments (optional):

I thank the authors for addressing the reviewers' feedback. They have addressed all of the points raised satisfactorily, so I am pleased to recommend this manuscript for publication in PLOS One.

Reviewers' comments:

Reviewer's Responses to Questions

**Comments to the Author**

Reviewer #3: All comments have been addressed

2. Is the manuscript technically sound, and do the data support the conclusions?

Reviewer #3: Yes

3. Has the statistical analysis been performed appropriately and rigorously?

Reviewer #3: Yes

4. Have the authors made all data underlying the findings in their manuscript fully available?

Reviewer #3: Yes

5. Is the manuscript presented in an intelligible fashion and written in standard English?

Reviewer #3: Yes

Reviewer #3: The authors have addressed most of my concerns and some were put into future work section. I hope they preform the future work soon.

**Do you want your identity to be public for this peer review?** For information about this choice, including consent withdrawal, please see our Privacy Policy

Reviewer #3: No

---

## [Editor Report · Acceptance letter]

PONE-D-24-10717R2

PLOS ONE

Dear Dr. Ali,

I'm pleased to inform you that your manuscript has been deemed suitable for publication in PLOS ONE. Congratulations! Your manuscript is now being handed over to our production team.

Kind regards,

on behalf of

Dr. Laura Morett

Academic Editor

PLOS ONE